# Clinical Characteristics of Acute Hepatitis E and Their Correlation with HEV Genotype 3 Subtypes in Italy

**DOI:** 10.3390/pathogens9100832

**Published:** 2020-10-11

**Authors:** Claudia Minosse, Elisa Biliotti, Daniele Lapa, Alessia Rianda, Mauro Marchili, Ilaria Luzzitelli, Maria Rosaria Capobianchi, Fiona McPhee, Anna Rosa Garbuglia, Gianpiero D’Offizi

**Affiliations:** 1Laboratory of Virology, “Lazzaro Spallanzani” National Institute for Infectious Diseases, IRCCS, 00149 Rome, Italy; claudia.minosse@inmi.it (C.M.); daniele.lapa@inmi.it (D.L.); maria.capobianchi@inmi.it (M.R.C.); 2Hepatology and Infectious Diseases Unit, “Lazzaro Spallanzani” National Institute for Infectious Diseases IRCCS, 00149 Rome, Italy; elisa.biliotti@inmi.it (E.B.); alessia.rianda@inmi.it (A.R.); ilaria.luzzitelli@inmi.it (I.L.); gianpiero.doffizi@inmi.it (G.D.); 3Ospedale San Camillo de Lellis, Hepathology Unit, 02100 Rieti, Italy; m.marchili@asl.rieti.it; 4Bristol-Myers Squibb Research and Development, Cambridge, MA 02142, USA; Fiona.mcphee@bms.com

**Keywords:** hepatitis E virus, molecular characterization, genotype 3, subtype, acute viral hepatitis

## Abstract

Genotype 3 (GT3) is responsible for most European autochthonous hepatitis E virus (HEV) infections. This study analyzed circulating genotypes and GT3 subtypes in the Lazio region, Italy, between 2011 and 2019, as well as their pathogenic characteristics. Of the 64 evaluable HEV GT3 patient-derived sequences, identified subtypes included GT3f (*n* = 36), GT3e (*n* = 15), GT3c (*n* = 9), GT3a (*n* = 1) and three unsubtyped GT3 sequences. GT3c strains were similar to Dutch sequences (96.8–98.1% identity), GT3e strains showed high similarity (96.8%) with a United Kingdom sequence, while the most related sequences to GT3f Italian strains were isolated in France, Belgium and Japan. One sequence was closely related to another Italian strain isolated in raw sewage in 2016. The liver functioning test median values for 56 evaluable GT3 patients were: alanine aminotransferase (ALT), 461 (range 52–4835 U/L); aspartate aminotransferase (AST), 659 (range 64–6588 U/L); and total bilirubin, 3.49 (range 0.4–33 mg/dL). The median HEV RNA viral load for 26 evaluable GT3 patients was 42,240 IU/mL (range 5680–895,490 IU/mL). Of the 37 GT3 patients with available clinical information, no correlation was observed between HEV clinical manifestations and GT3 subtype. HEV symptoms were comparable among GT3c/e/f patients across most analyzed categories except for epigastric pain, which occurred more frequently in patients with HEV GT3e (75%) than in patients with GT3c (50%) or GT3f (19%) (*p* = 0.01). Additionally, patients with HEV GT3c exhibited significantly higher median international normalized ratio (INR) than patients with GT3e and GT3f (*p* = 0.033). The severity of GT3 acute hepatitis E was not linked to HEV RNA viral load or to the GT3 subtype.

## 1. Introduction

Hepatitis E virus (HEV) is a non-enveloped, single-stranded, positive-sense RNA virus that is a member of the genus *Orthohepevirus* in the family Hepeviridae. There are currently five identified species (A–E) of *Orthohepevirus*, with species A having eight assigned genotypes (GTs), six of which (GT1–4, GT7 and GT8) can infect humans [1]. Genotypes 1 and 2 only infect humans and are endemic in low-income countries including Asia (GT1), Africa (GT1 and GT2) and South America (GT2). The other genotypes are transmitted zoonotically from animal reservoirs to humans mainly through contaminated food. GT3 has been detected worldwide in both domestic mammals such as pigs, rabbits and sheep and wild mammals including wild boar and deer [2,3]. GT4 has been detected in similar human and mammalian reservoirs to GT3, but is found mainly in Southeast Asia and India with only sporadic autochthonous cases being described in Europe [4,5]. Genotypes 7 and 8 have been detected in the Middle East [6,7].

HEV is considered as the main causative agent of enterically transmitted hepatitis in the world [8,9]. It is responsible for approximately 50% of acute hepatitis in low-income countries and according to the World Health Organization (WHO), one third of the population has been exposed to HEV [9].

In Europe, it has been estimated that over 2 million local HEV-acquired infections occur annually [10,11]. In Germany, HEV cases have increased 40-fold over a 10-year period (2002–2012) [12]. In France, 2000 laboratory-confirmed HEV infections have been attributed to GT3 with sporadic acute cases being attributed to GT4 [13]. The recent increase in reported acute infections is due to greater awareness among physicians and the surveillance system implemented in many European countries [10].

Overall, GT3 is responsible for the majority of the autochthonous cases in Europe [14]. Seroprevalence in European countries ranges from 5% [15,16] to 50% in some regions of Southern France [17]. Differences in seroprevalence may be a consequence of the sensitivity of the anti-HEV antibody detection assay employed [18].

GT3 and GT4 HEV infections are rarely associated with clinical symptoms, although fulminant hepatitis disease has been reported for 0.4–0.5% of cases [19]. Chronic hepatitis has been observed in immunosuppressed patients [13]. Data reported after a HEV outbreak showed that approximatively 70% of people who contracted the virus due to consumption of undercooked meat were asymptomatic [20,21].

For GT3 or GT4 HEV infections resulting in clinical symptoms, elevation of liver enzymes, jaundice and other non-specific symptoms like fever, nausea, vomiting, malaise/fatigue or anorexia have been reported. Dark urine and jaundice signal the onset of the icteric phase of the disease. Acute hepatitis E resolves spontaneously in most cases after 1–2 weeks with a normalization of liver enzymes (i.e., ALT and AST), and only in rare cases, complications such as agranulocytosis, progressive bilirubinemia and neutropenia can persist after acute hepatitis resolution [22]. Clinical manifestations are observed mainly in males over 60 years old [23].

Even though HEV infection is subclinical in most patients, it represents the main cause of acute hepatitis in several European countries. Moreover, in England, France and Germany, where a hepatitis surveillance system has been employed, HEV cases exceeded hepatitis A virus (HAV) cases [10].

Our understanding of factors affecting the severity of an acute HEV infection are still evolving. The aim of this study was to analyze the genotype and GT3 subtypes circulating in the Lazio region in Italy during a 9-year period (2011–2019) and their pathogenic characteristics in human HEV infections.

## 2. Results

### 2.1. Acute Hepatitis E Diagnosed between 2011 and 2019

Of the 3443 patients tested for anti-HEV IgM and IgG during 2011–2019, 295 (8.6%, 295/3443) were positive for both serological markers. The median age of anti-HEV IgM positive patients was 53 (range 18–85 years). HEV RNA was detected in 88 (34.4%) of 256 available anti-HEV IgG and IgM positive patient-derived samples (Figure 1). The open reading frame 2 (ORF2) region was amplified from 77 samples, while ORF1 was amplified from 11 samples. The sequences were aligned with standard reference viral strains representing published genotypes/subtypes and confirmed by phylogenetic analysis (see Section 2.2 below). Fourteen sequences aligned with GT1, 5 with GT4 [4] and 69 with GT3 (Figure 1). Sanger sequencing only led to the resolution of short fragments; hence, 5 of the 69 GT3 sequences were excluded from the phylogenetic analysis. GT3 sequences were grouped by five interval year periods: 2011–2013, 2014–2016, 2017, 2018 and 2019 due to the limited sample size. GT3f was the prevalent subtype up to 2018. In 2019, an increase of GT3e strains resulted, a consequence of an HEV outbreak during the summer. These GT3e patients harbored the same strains (99.5% identity); all lived in proximal towns and had consumed undercooked pork, probably prepared in the same farm factory (unpublished data, epidemiological investigation in process). No patient had been in contact with clinical cases, confirming rare interpersonal transmission of HEV GT3. The distribution of the 64 evaluable GT3 subtypes is shown in Figure 2. The median age of these GT3 patients was 61.2 years (range 30–85 years) and 87.5% were male.

ALT, AST and total bilirubin values were available for 56 of 64 GT3 patients, HEV RNA viral load values were available for 26 patients (Venn diagram, Figure 3) and clinical data were available for 37 patients (see Section 2.3). Overall, in GT3 patients (*n* = 64), the median values for ALT, AST and total bilirubin were 461 U/L (range 52–4835 U/L), 659 U/L (range 64–6588 U/L) and 3.49 mg/dL (range 0.4–33 U/L), respectively. The median HEV RNA viral load was 42,240 IU/mL (range 5680–895,490 IU/mL). A higher viral load was associated with the GT3e subtype (median 59,820 IU/mL; range 14,210–895,490 IU/mL) when compared with GT3f (median 35,600 IU/mL; range 14,140–172,350 IU/mL) and GT3c (median 14,560 IU/mL; range 5680–23,440 IU/mL). However, statistical significance was not established (*p* > 0.05, Figure 4A) among subtypes, a possible consequence of the limited sample size. Additionally, no correlation was determined between the evaluated clinical characteristics (ALT, AST and total bilirubin) and GT3 subtypes (Figure 4A). Moreover, no correlation was established between patient age and clinical characteristics or viral load (Figure 4B).

### 2.2. Genotype and Phylogenetic Analysis

Genotype classification of 64 HEV GT3 patient-derived sequences (53 targeting the ORF2 region and 11 targeting the ORF1 region) was first accomplished by performing a sequence similarity BLAST search. To further characterize our GT3 sequences, a phylogenetic analysis was also performed according Smith et al. [24,25].

For the 53 GT3 ORF2 sequences, the analysis was performed by including HEV reference sequences and 97 globally circulating GT3 public database sequences. Most sequences (*n* = 44) clustered with clade 3efg: 29 were GT3f and 15 were GT3e. Nine sequences clustered with clade 3abkchij: 7 were GT3c, 1 was a previously described GT3a sequence (INMI_1736_2017) [26] and 1 strain was unsubtyped (INMI_1909_2019) (Figure 5A). No GT3b, GT3g, GT3h, GT3k, GT3i or GT3j subtypes were identified among study patient-derived sequences.

Among GT3f sequences, 21 out of 29 sequences were bucketed into five subclusters (A–E, Figure 5B). In subcluster A, the p-distance was 0.02271 (SE 0.00563). All three strains were isolated in 2017, but no common origin of source infection was identified. The most closely related strain was isolated from a patient in Japan in 2015 (LC192453), demonstrating 98.5% similarity with our INMI_1715_2017 strain. In subcluster B, the p-distance was 0.00097 (SE 0.00098). All five strains were collected in 2017 and presumably, the source of infection was the same for all patients. A French strain, KC166968, exhibited the highest sequence identity with GT3f subclusters B (96.3%) and A (96.1%). Subcluster C included seven strains collected in 2019 with a p-distance of 0.00487 (SE 0.00199) and a similarity of 97.3% with a French sequence MN646689 which belonged to a blood donor identified in 2004. The two sequences in subcluster D isolated in 2012 and 2015, respectively, had a p-distance of 0.02920 (SE 0.00793) and some similarity (93.2%) with a Belgium sequence (MN614143, 2014). The four strains in subcluster E showed a p-distance of 0.01135 (SE 0.00365). Strain INMI_1823_2018 exhibited a similarity of 95.1% with the KF891380 sequence isolated from a swine in Emilia Romagna, a region of Central Italy (2013) and strain INMI_1609_2016 also exhibited similar identity (94.4%).

Among the GT3e subtype strains, 13 out of 15 strains were split into two subclusters. In subcluster A, four sequences (p-distance 0.01987, SE 0.00511) were identified that were isolated from different years. Strains INMI_1604_2016 and INMI_1611_2016 showed high similarity (96.8%) with a sequence from the United Kingdom (MK167982). This sequence also had high similarity with the nine sequences included in subcluster B (similarity 97.3%). Subcluster B strains had a p-distance of 0.00054 (SE 0.00049) and were isolated during July–September 2019, suggesting a common source of infection.

There were no specific subclusters for the seven GT3c strains (p-distance 0.04639, SE 0.00676). High identity was shown between these strains and several sequences originating from the Netherlands (Figure 5D), although the highest similarity was observed between INMI_1901_2019 and a French sequence, MF444115 (similarity 99.3%). Four sequences were similar to Dutch sequences (identity range: 96.8–98.1%).

Other study strains were interspersed along the tree showing similarity with sequences from different countries (France, Singapore, Sweden, United Kingdom, Italy, Germany and Spain) (Figure 5B–D).

The phylogenetic analysis of 11 strains identified by amplifying ORF1 sequences is presented in Figure 6. Ten sequences clustered with GT3f (*n* = 7), GT3c (*n* = 1) or GT3e (*n* = 2). Sequences were interspersed along the neighbor-joining tree with only two sequences (INMI_1411_2014 and INMI 1625_2015) being in the same cluster (p-distance 0.02, SE 0.01). One sequence (INMI_1804_2018) was closely related to an Italian strain isolated in raw sewage in 2016 (MN251650) (Figure 6).

### 2.3. Clinical Manifestations in HEV GT3 Acute Hepatitis

Of the 88 patients with HEV acute hepatitis, clinical information was available for 37 with HEV GT3. Baseline demographic characteristics and underlying comorbidities are reported in Table 1. The median age was 59 years (range, 52–68 years) and the majority (86.5%) were males. Overall, 40.7% of patients reported alcohol intake, 30.8% were smokers and 5.4% were people who injected drugs (PWID). A high proportion of patients had at least one underlying comorbidity; 8.1% had chronic liver disease and 73% had an extra-hepatic comorbidity. The most frequent comorbidities were arterial hypertension (45.9%), diabetes (35.1%), cardiovascular diseases (21.6%) and lipid disorders (18.9%).

Clinical characteristics and laboratory results of HEV GT3 study patients are reported in Table 2. Almost half (*n* = 15, 40.5%) of these patients developed icteric hepatitis. The clinical presentation of HEV infection included asthenia (*n* = 18, 48.6%), fever (*n* = 12, 32.4%), nausea or vomiting (40.5%) and epigastric pain (35.1%). Symptoms were experienced for a median of 7 (5–9.5) days before hospital admission and patients were hospitalized for a median of 9 (7–11) days. The median bilirubin levels at peak were 7 (2.3–12.2) mg/dL, while the median ALT values and international normalized ratio (INR) were 1751 (1026–2810) U/L and 1.17 (1.08–1.5), respectively. Overall, 13 (36.1%) patients had severe acute hepatitis and two developed liver decompensation with ascites. No patient died within three months of disease onset.

Patients who developed severe HEV acute hepatitis had similar demographic characteristics and underlying comorbidities compared with those who did not (Appendix A), and, as expected, they showed a significantly longer duration of hospitalization (11 (9.5–21.5) days vs. 8 (6.2–9), *p* = 0.001) (Appendix A. Appendix A).

In general, acute HEV GT3-infected patients with different subtypes (3c, *n* = 6; 3e, *n* = 8; and 3f, *n* = 21) were comparable in terms of age, sex, body mass index (BMI), alcohol consumption, smoking, pre-existing chronic liver disease, number and type of comorbidities, as well as duration of both symptoms and hospitalization (Table 1 and Table 2). Interestingly, symptoms were comparable among patients with different GT3 subtypes, except for epigastric pain which occurred more frequently in patients infected with HEV GT3e (75%) than in patients infected with GT3c (50%) or GT3f (19%, *p* = 0.01). Laboratory results (AST, ALT, total bilirubin, γ glutamil transferase (γGT) and alkaline phoshatase (ALP) were similar for patients irrespective of the infecting HEV GT3 subtype, although patients with GT3c had a significantly higher median INR than patients with GT3e and GT3f (1.6 vs. 1.15 and 1.15, *p* = 0.033). The outcome of acute hepatitis was similar among the different HEV GT3 subtypes; however, severe acute hepatitis tended to develop less frequently (12.5%) in patients with GT3e compared to patients with GT3c (50%) or GT3f (42.8%), although the difference was not statistically significant.

A comparison of clinical characteristics of patients infected with HEV GT3e subcluster B and GT3f subcluster D was possible. For GT3e subcluster B, clinical characteristics were available for six patients. Median age was 59 years (range 56–64 years), 83.3% were males and BMI was 24.3 ± 3.1 kg/m^2^. Three patients (50%) had at least one extra-hepatic comorbidity, two patients had one comorbidity (arterial hypertension in one case and diabetes in the other case) and one had three comorbidities (arterial hypertension, cardiovascular disease and respiratory diseases). The clinical presentation of HEV infection included epigastric pain (83.3%), asthenia (66.7%), nausea or vomiting (50%) and arthralgia (50%). The median duration of symptoms before hospitalization was 11 (7.2–15) days. Median bilirubin peak levels (2.5 (1.0–5.1) mg/dL) and ALT values (1395.5 (335–2300) U/L) tended to be lower compared with those reported in the study population, while the median INR (1.15 (1.01–1.3)) was comparable. The median hospitalization duration was 7.5 (6.5–9) days. None of these patients developed severe acute hepatitis, and no specific comorbidity was linked to more elevated ALT, AST and/or total bilirubin values.

For GT3f subcluster C, clinical characteristics were available for five patients. The median age was 71 years (range 68–75 years), 100% were males and BMI was 23.9 ± 4.8 kg/m^2^. Four patients (80%) had at least one extra-hepatic comorbidity: two patients had one comorbidity (arterial hypertension in one case and cardiovascular disease in the other case), one had three comorbidities (arterial hypertension, cardiovascular disease and renal disease) and one had five comorbidities (arterial hypertension, diabetes, cardiovascular disease, lipid disorders and neoplastic disease). The clinical presentation of HEV infection included fever (40%), asthenia (20%), nausea or vomiting (20%) and diarrhea (20%). The median duration of symptoms before hospitalization was 6 (3–10.5) days. Median bilirubin peak levels (3.2 (1.2–14.8) mg/dL) were less than those reported in the study, while median ALT values (1859 (906–3578.5) U/L) and the median INR (1.1 (1.1–1.4)) were comparable. The median hospitalization duration was 7 (5–13.5) days. For the single patient who developed severe acute hepatitis, no specific comorbidity was apparent.

## 3. Discussion

In this study, the occurrence of acute HEV GT3 infections was surveyed in the Lazio region of Italy between January 2011 and December 2019. The influence of GT3 subtypes and genetic variability of viral strains on the course and severity of clinical manifestations was also evaluated in HEV RNA-positive patients. GT3f was the most frequently observed subtype between 2011 and 2018, with a prevalence ranging from 36.4% to 73.3%. The frequency of GT3e subtypes ranged from 6.7–45.5% (Figure 2). The prevalence of HEV GT3e acute infections was high in 2019, attributable to an outbreak in people vacationing in the Abruzzo region. GT3c represented the third most abundant subtype (9 cases), while only a single GT3a case [26] was observed. Our GT3f patient-derived sequences were most similar to French human sequences, KC166968 (96.1% with INMI_1719_2017) and MN646689 (97.3%); a Belgian human sequence, MN614143 (93.2%); and one Italian pig sequence, KF891380 (94.4%). One of our GT3f sequences (INMI_1715_2017) revealed high similarity (98.5%) with a non-European strain: a Japanese GT3f human sequence (LC192453) [27]. Patient INMI_1715_2017 had not traveled abroad during six months prior to the onset of clinical symptoms; thus, it is more likely that the Japanese patient ingested imported European pork [27].

Our GT3e patient-derived study sequences exhibited high similarity (cluster A, 97.3%; cluster B, 96.8%) with a sequence from the United Kingdom, MK167982 (human, 2015). Five of our seven GT3c sequences showed >96% similarity with Dutch sequences. One French human sequence, MF444115 (2015), exhibited a similarity of 99.3% with INMI_1901_2019. These data suggest that in the human host, there may be a selection of variants harbored in swine viral quasispecies. However, in the absence of ultra-deep sequencing analyses of viral sequences from the contaminating animal source and the human host, this hypothesis cannot be proven. Given the high similarity observed between strains isolated from European countries, it is likely that the source of contamination originated from imported meat distributed by local supermarket chains or by pigs purchased abroad that were farmed and slaughtered in Italy. In swine, HEV has an R_0_ value of 8.8 [28,29], therefore, one animal can transmit the infection to an entire herd. Thus, viral strains can spread rapidly to different geographic areas.

It is interesting that there was high similarity (98.4%) between study sequence INMI1804_2018 and MN251650, a sequence identified in Italian wastewater [30]. This finding reinforces evidence, already described by other countries, that inadequate management of sewage treatment plants could represent a critical factor for environmental HEV transmission, including on cultivated land [31,32].

An analysis of the various clusters within genotype subtypes GT3f and GT3e exposed how the severity of biochemical parameters do not depend on the viral strain. In GT3f subtype cluster C, for example, ALT values varied from 33 to 4011 U/L, AST ranged from 99 to 2859 U/L and total bilirubin values varied from 0.5 to 18.68 mg/dL. In a patient with severe hepatitis, ALT, AST and total bilirubin values were lower than values observed in patients with mild, acute hepatitis. Overall, ALT, AST and total bilirubin values were 1.5–2 times higher in HEV-infected patients with GT3e compared to those with GT3f and GT3c. Intriguingly, these higher biochemical values did not correspond with greater severity in clinical symptoms.

In GT3e subtype cluster B, transaminase values varied widely from 73 to 2114 U/L for ALT, 38 to 2787 U/L for AST and 0.6 to 13.6 mg/dL for total bilirubin. However, there was high similarity (99.5%) between the nine isolated patient viral strains in this cluster.

An analysis of clinical symptoms revealed that the severity of certain clinical symptoms was associated with specific GT3 subtypes. For example, epigastric pain was observed more frequently in patients infected with the HEV GT3e subtype than in patients infected with GT3c and GT3f (*p* = 0.01). Conversely, median INR was higher in patients infected with the HEV GT3c subtype than in patients infected with GT3e and GT3f (*p* = 0.033) (Table 2). As expected, patients with severe hepatitis had higher bilirubin values than those with mild hepatitis (Appendix A). Patients with chronic liver disease and PWID were more likely to have severe acute hepatitis E (Appendix A). However, statistical significance was not established, perhaps a consequence of the limited study sample size. Overall, acute hepatitis disease severity was not associated with viral strains representing specific GT3 subtype clusters.

The median viral load in our study was lower than that observed in symptomatic French patients described by Lhomme et al. (42,240 IU/mL vs. 282,000 IU/mL) [33], and more comparable to values described in asymptomatic patients from Germany and France [18,34]. Additionally, a correlation of viral load with severity of acute hepatitis or biochemical parameters was not observed in our study.

The higher viral load in symptomatic French patients may be explained by the viral titer present at the source of HEV transmission. Notoriously, “figatelli”, a traditional French sausage based on pork liver, or pork paté can be consumed without cooking; therefore, the HEV titer is high when transmitted to humans [35,36]. In Italy, pork is generally roasted before eating and dried sausages are rarely consumed raw. Barbequed pork could be undercooked; however, the infectious viral titer should be lower than that in figatelli or paté [36,37,38]. In interviews carried out at the hepatology Unit, only two patients in our study admitted to eating dried sausage. Most patients had ingested undercooked pork with presumably a low titer of infectious virions. This could explain the lower viral loads detected in patients during the acute phase of hepatitis.

Our study has several limitations. Firstly, clinical information and virological parameters (ALT, AST, total bilirubin, HEV viral load and clinical manifestations), required to evaluate the severity of acute hepatitis, were not available for all patients. This was because only a few samples sent to the Laboratory of Virology, National Institute of Infectious Diseases (INMI) L Spallanzani, for HEV testing had accompanying clinical information.

This weakened the power of statistical analyses; therefore, categories where statistical significance was observed among GT3 subtypes in our study must be considered with caution.

A correlation between GT3 subtypes and pathogenicity of acute hepatitis E disease was not observed, perhaps a consequence of the limited datasets. Moreover, it was difficult to compare our findings with data reported in the literature from other European countries, since prior reports did not evaluate symptomatic and asymptomatic patients separately [39] or comorbidities were not taken into account [40]. Italian zoonotic strains have often been genotyped by sequencing non-overlapping regions with our sequence target; thus, a comparison between human and animal HEV sequences was limited and was considered to be preliminary in our study.

Our results suggest that GT3 acute hepatitis severity is not linked to the infecting HEV GT3 subtype viral load. Only select clinical symptoms were more frequently observed in patients with specific GT3 subtypes.

## 4. Methods

### 4.1. Study Design

Between January 2011 and December 2019, 3443 patients were tested for anti-HEV IgM and anti-HEV IgG antibodies. Patients included those who were hospitalized and outpatients. All had abnormal ALT levels and were negative for hepatitis B virus (HBV), HAV and hepatitis C virus (HCV) genomic and serological markers. HEV antibody detection was performed using an enzyme-linked immunoassay (ELISA) (DIA.PRO, Milan, Italy). All anti-HEV IgM-positive patients were tested for HEV RNA. Samples positive for HEV RNA were genotyped. Inclusion criteria for defining HEV acute hepatitis included >2.5-fold elevation of normal alanine transaminase levels and/or bilirubin >1 mg/mL in serum. Liver decompensation was defined by the acute development of one or more major liver complications including ascites, hepatic encephalopathy, gastrointestinal hemorrhage or bacterial infections. Severe acute hepatitis was defined by the presence of an international normalized ratio (INR) ≥ 1.5 and/or a total bilirubin level higher than ten times the upper limit of normal (ULN), as previously published [41]. The ULN for total bilirubin levels was 1 mg/dL, according to our laboratory reference range. A detailed record of travel history, food, blood transfusion and clinical symptoms was collected from patients admitted to the Hepathology Unit. This study was approved by INMI L Spallanzani Istituto di ricovero e cura a carattere scientifico (IRCCS) Hospital Ethics Committee (agreement 29/2013, approved on 05-21-2013).

### 4.2. HEV RNA Detection and Sequencing

Serum samples from 256 out of 295 anti-HEV IgM-positive patients were submitted for HEV RNA testing. RNA was extracted from 400 µL of serum or plasma using the QIASYMPHONY automated instrument (QIAGEN, Hilden, Germany). In samples collected from January 2011 to December 2018, reverse transcription and first round amplification were performed by one-step RT-PCR using ORF2 primers. Resultant product (5 µL) was used in a second round of PCR, as previously described [26]. From January 2019 onwards, a more sensitive detection protocol was employed. cDNA was retro-transcribed using the SUPERSCRIPT IV reverse transcriptase (SSIV) first round system (Thermo Fisher Scientific, Paisley, United Kingdom) with hexamer random primers according to the manufacturer’s instructions. A nested PCR technique was used to amplify DNA fragments in the ORF2 gene. The first round of PCR was performed as previously described (De Sabato Frontiers Microbiol 2020) using 10 µL cDNA in a total mixture of 50 µL containing 10x buffer, 2.5 µL MgCl_2_ (20 mM MgCl_2_ stock solution), 10 mM dNTP, 2.5.IU TaqGold polymerase (Applied Biosystem, Forster City, CA, USA) and 0.5 µL each of forward and reverse outer primers (50 µM) (Table 3). The same conditions were employed for the second round of PCR, except 5 µL of first round product was used as template and forward and reverse nested primers were used (Table 3). If no HEV RNA product was detected with ORF2 primers, samples were amplified using primers representing a conserved ORF1 region [42]. RNA was transcribed in the presence of random hexamers and SSIV. cDNA (10 µL) was used as a template in the first round of PCR with outer primers 1679 and 1680, while nested primers 1681 and 1682 were employed for the second round of PCR (Table 3). The cycling conditions for the first round of PCR were: 94 °C, 15 min; 40 cycles at 94 °C for 30 sec, 50 °C for 45 sec and 72 °C for 1 min; and a final extension at 72 °C for 7 min. The reaction mixture was prepared using TaqGold polymerase and the same reagents employed for the ORF2 amplification. Positive amplicons were purified using the QIAQUICK PCR product (QIAGEN, Hilden, Germany) and bi-directionally sequenced using the Big Dye Terminator v3.1 kit and BI 3130 autosequencer (Applied Biosystems, Forster City, CA, USA). The HEV FASTA sequences were compared using GenBank nucleotide BLAST [43] and were typed according to Smith et al. [24,25]. Primer sequences were removed before the comparison analysis.

### 4.3. Phylogenetic Analysis

A phylogenetic tree comprised of sequences representing the ORF2 region (411 base pair, bp) was constructed using the maximum likelihood method based on the general time reversible model + G using MEGAv.10 software (www.megasofware.net). Confidence values of internal nodes were calculated by performing a 1000 bootstrap analysis including reference sequences [24,25] and sequences with a strict similarity with study sequences. GT3 sequences were subgrouped into three major clades: 3efg, 3abchij and 3ra [24,25].

A second phylogenetic tree comprised of sequences representing part of the ORF1 region (132 bp) was constructed employing the maximum likelihood method with the Kimura 2-parameter model + G + I. Even though the ORF1 fragment was small and not designated as a primary target choice for HEV subtyping, a subtype for our patient-derived sequences was only assigned when >98% similarity was observed with a reference sequence. The robustness of phylogenies was tested using bootstrap analysis [44].

All amplified patient-derived sequences from this study are deposited in GenBank (accession numbers: ORF2: MN444828–MN444850, MN444852, MN444853, MN604404, MN737482–MN737484, MT769324, MT769325, MT263583–MT263585, MN509469–MN509471, MN497623, MN537875–MN537879, MT769321–MT769325, MT786653, MT763865–MT763867; ORF1: MT763857–MT763864, MT769326, MT707659, MT731331)**.**

### 4.4. HEV RNA Quantification

Residual RNA extracted by QIASYMPHONY was stored at −80 °C. RNA samples were re-analyzed later in January 2020 using the Real Star HEV RNA kit (Altona diagnostics GmbH, Hamburg, Germany). For HEV RNA quantification, serial dilutions of 10,000, 1000, 100, 50, 25, 12.5 and 6.25 IU/mL from a HEV RNA standard (1st International Standard WHO, code 6329/10) were prepared. All procedures were performed according to the manufacturer’s instructions. The lower limit of detection (LOD) was 10 IU/mL. Data were analyzed using Rotorgene software, version 2.1.0. When the Threshold Cycle (Ct) of a sample internal control was at least three cycles higher than the Ct of the non-template internal control, it was excluded from analysis.

### 4.5. Statistical Analysis

Continuous variables were summarized as mean ± standard deviation or median ± interquartile range (IQR), while categorical data were expressed as counts and percentages. Comparisons between groups were performed using the chi-square test or Fisher’s exact test for categorical variables or the Kruskal–Wallis test or Mann–Whitney test for continuous variables. The significance level for all analyses was set at *p* < 0.05.

## Figures and Tables

**Figure 1 pathogens-09-00832-f001:**
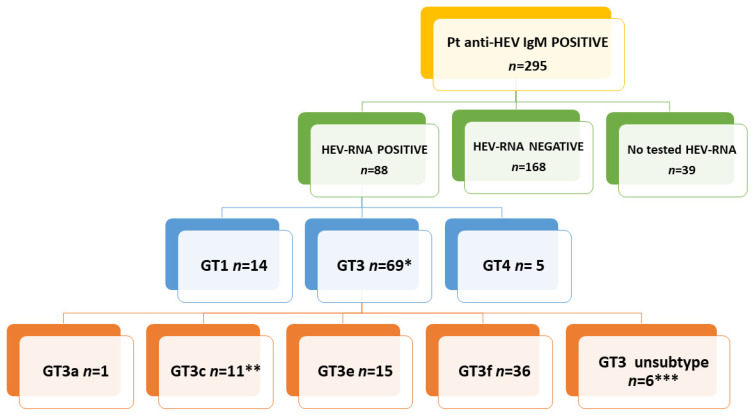
Flow chart of anti-hepatitis E virus (HEV) IgM positive patients, HEV RNA positivity and genotype 3 (GT3) subtype determination. *, five GT3 sequences were excluded from the phylogenetic analysis because sequenced fragments were too short; **, two GT3c sequences were excluded from the phylogenetic analysis because sequences were too short; ***, three GT3 unsubtyped sequences were excluded from the phylogenetic analysis because sequenced fragments were too short.

**Figure 2 pathogens-09-00832-f002:**
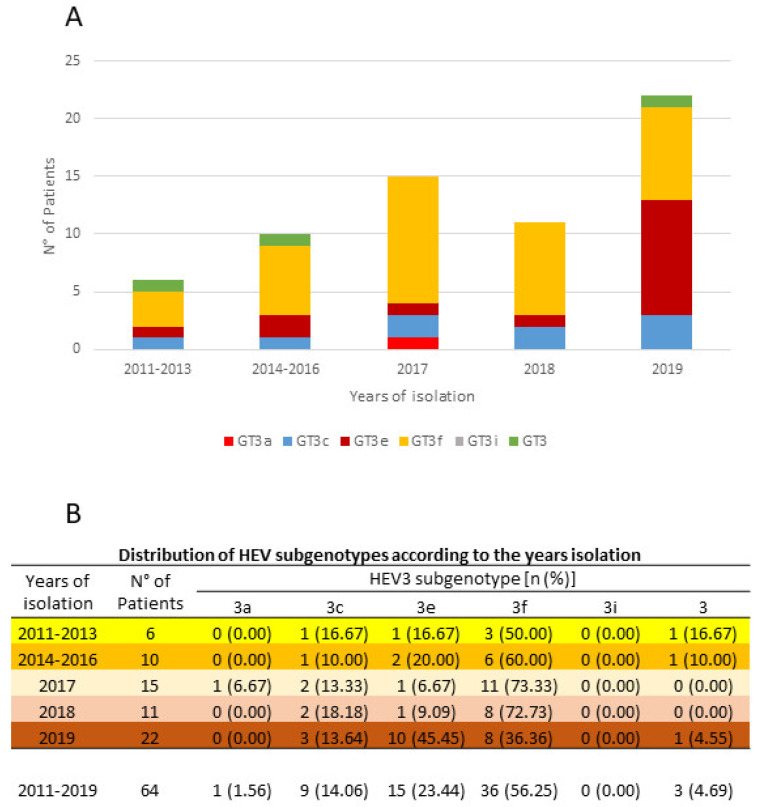
Distribution of isolated GT3 subtypes (**A**) and prevalence of each GT3 subtype (**B**) by year interval.

**Figure 3 pathogens-09-00832-f003:**
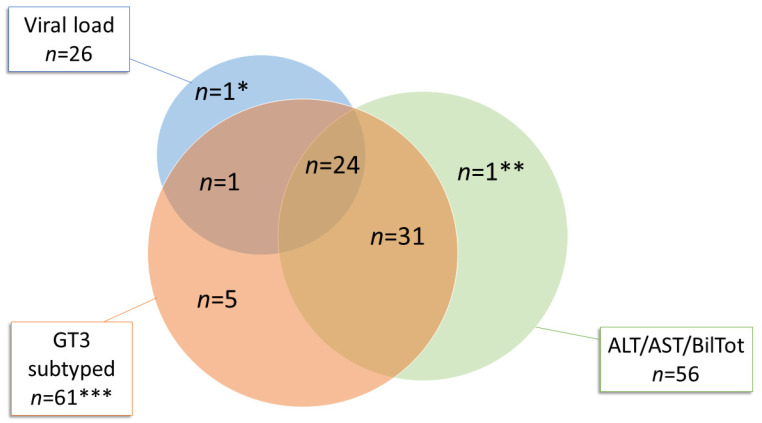
Venn diagram. Representation of available subtyped GT3 strains, HEV RNA viral load and liver function test results (ALT, AST and total bilirubin). * HEV GT3 unsubtyped patient whose ALT, AST and total bilirubin values were not available. ** HEV GT3 unsubtyped patient with unavailable HEV RNA viral load. *** Three out of sixty-four HEV GT3 sequences were unsubtyped: for one patient, only HEV RNA viral load was available (*); for another patient only values of ALT, AST and total bilirubin were available (**); and for the third GT3 unsubtyped patient (excluded from the Venn diagram), neither HEV RNA viral load nor ALT, AST and total bilirubin values were available.

**Figure 4 pathogens-09-00832-f004:**
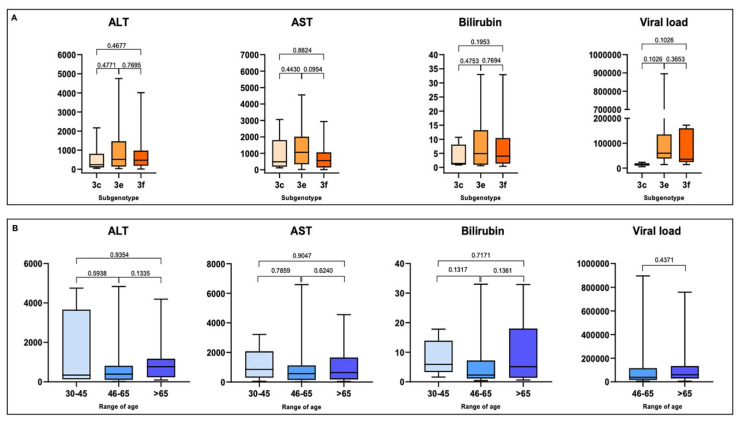
(**A**) Correlation between GT3 subtypes (*x*-axis) and ALT (U/L), AST (U/L), total bilirubin (mg/dL) and HEV RNA viral load (IU/mL) (*y*-axis). ALT, AST and total bilirubin were analyzed in 55 GT3 patients (3c, *n* = 9; 3e, *n* = 14; 3f, *n* = 32). HEV RNA viral load was considered for 24 GT3 subtyped patients (3c, *n* = 2; 3e, *n* = 11; 3f, *n* = 11), one GT3a sample was excluded from the analysis. (**B**) Correlation between age (*x*-axis) and ALT (U/L), AST (U/L), total bilirubin (mg/dL) and HEV RNA viral load (IU/mL) (*y*-axis). ALT, AST and total bilirubin were analyzed in 55 GT3 subtyped patients (age range 30–45 years, *n* = 5; 46–65 years, *n* = 31; >65 years, *n* = 19). HEV RNA viral load was available for 25 GT3 subtyped patients (age range 30–45 years, *n* = 0; 46–65 years, *n* = 12; >65 years, *n* = 13). *p*-Values between studied parameters are provided above the respective graph columns.

**Figure 5 pathogens-09-00832-f005:**
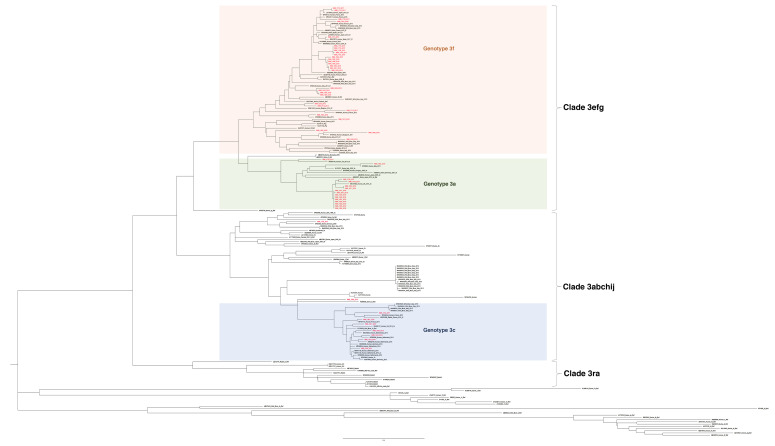
Phylogenetic analysis of open reading frame 2 (ORF2) (411 nucleotides, nt) including patient-derived study sequences (in bold), reference sequences [24] and 97 GT3 sequences. Study samples are indicated by isolate name. The GenBank accession number, location, host and date of isolation, if applicable, are specified for other sequences. The scale bar indicates nucleotide substitution per site. Bootstrap values >70% are indicated. Panel (**a**) all analyzed ORF2 GT3 sequences by clade; (**b**) GT3f sequences; (**c**) GT3e sequences; (**d**) GT3c sequences. The scale bar represents nucleotide substitutions per site.

**Figure 6 pathogens-09-00832-f006:**
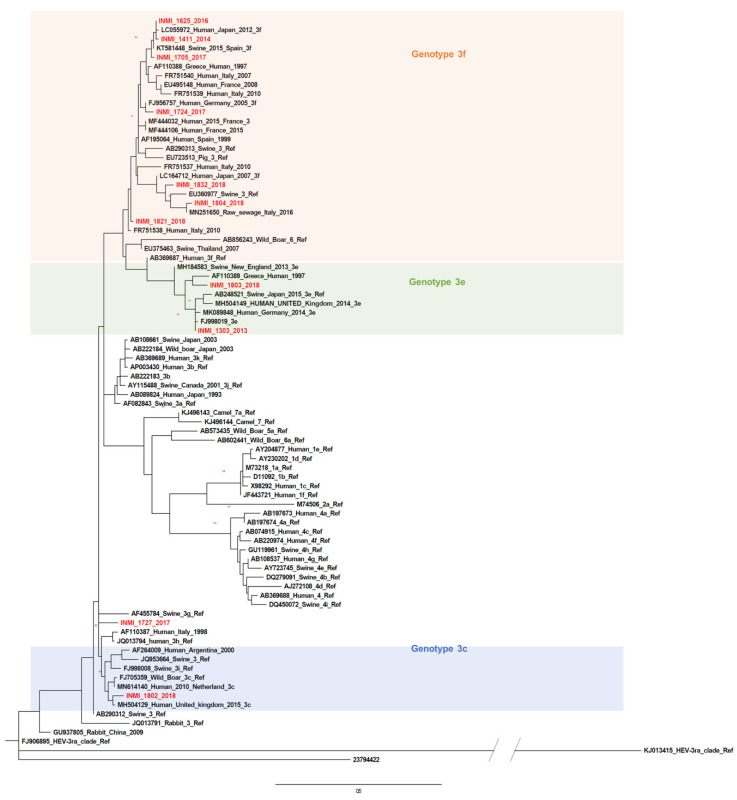
Phylogenetic analysis based on partial open reading frame 1 (ORF1, 125 nt, nt 78-202) of the HE-JA04-1911 isolate (GenBank accession number AB248521) and patient-derived sequences obtained in this study. The phylogenetic tree includes the prototype strains as indicated by Smith et al. [24] and other sequences isolated from different geographical regions with a high percentage of similarity to our sequences. Previously described HEV GT3 subtypes are indicated. The GT1 sequence was used as the outgroup. The phylogenetic tree was built by using the maximum likelihood method based on the general time reversible model with a bootstrap of 500 replicates. Bootstrap values >60% are indicated on respective branches. The scale bar represents nucleotide substitutions per site.

**Table 1 pathogens-09-00832-t001:** Baseline demographic characteristics and underlying comorbidities in acute HEV-infected patients by HEV GT3 subtype.

Characteristics	HEV GT3	HEV GT3c	HEV GT3e	HEV GT3f	*p-*Value
	(*n* = 37)	(*n* = 6)	(*n* = 8)	(*n* = 21)	
**Age, years median (IQR)**	59 (52–68)	59.5 (51.5–63)	59 (49–66)	65 (53–71)	0.2
Male sex (*n*, %)	32 (86.5%)	6 (100%)	7 (87.5%)	17 (80.9%)	0.2
BMI, kg/m^2^	24.66 ± 3.13	27.9 ± 4.9	24.6 ± 2.9	24.0 ± 2.9	0.6
Alcohol user, (*n*, %)	11/27 (40.7%)	1/2 (50%)	1/7 (14.3%)	7/17 (41.2%)	0.4
Smoker, (*n*, %)	8/26 (30.8%)	1/2 (50%)	1/6 (16.7%)	5/16 (31.2%)	0.6
PWID, (*n*, %)	2/37 (5.4%)	0	0	2/21 (9.5%)	0.9
Chronic liver disease, (*n*, %)	3/37 (8.1%)	1 (16.7%)	0	2 (9.5%)	0.9
Number of comorbidities					
0	10 (27%)	1 (16.7%)	4 (50%)	3 (14.2%)	
1	15 (40.5%)	2 (33.3%)	2 (25%)	11 (52.4%)	
2	2 (5.4%)	1 (16.7%)	0	1 (4.7%)	
3	7 (18.9%)	2 (33.3%)	2 (25%)	3 (14.3%)	
4	1 (2.7%)	0	0	1 (4.7%)	
5	2 (5.4%)	0	0	2 (9.5%)	
Diabetes, (*n*, %)	13 (35.1%)	3 (50%)	2 (25%)	8 (38.1%)	0.6
Cardiovascular diseases, (*n*, %)	8 (21.6%)	1 (16.7%)	1 (12.5%)	6 (28.6%)	0.6
Lipid disorders, (*n*, %)	7 (18.9%)	1 (16.7%)	1 (12.5%)	5 (23.8%)	0.7
Renal diseases, (n, %)	1 (2.7%)	0	0	1 (4.8%)	
Arterial hypertension, (*n*, %)	17 (45.9%)	4 (66.7%)	3 (37.5%)	10 (47.6%)	0.5
Digestive diseases, (*n*, %)	3 (8.1%)	0	0	3 (14.3%)	0.9
Respiratory diseases, (*n*, %)	3 (8.1%)	1 (16.7%)	1 (12.5%)	1 (4.8%)	0.6
Neoplastic diseases, (*n*, %)	2 (5.4%)	0	0	2 (9.5%)	0.9

Interquartile range, IQR; body mass index, BMI; people who inject drugs, PWID.

**Table 2 pathogens-09-00832-t002:** Clinical characteristics and laboratory results for acute HEV-infected patients by HEV GT3 subtype.

Characteristics	HEV GT3 (*n* = 37)	HEV GT3c (*n* = 6)	HEV GT3e (*n* = 8)	HEV GT3f (*n* = 21)	*p*-Value
Duration of hospitalization, days	9 (7–11)	9.5 (8.5–13.25)	8 (7–8.75)	10 (6.5–16.5)	0.28
Duration of symptoms before hospitalization, days	7 (5–9.5)	5.5 (2.7–15)	9 (2.7–13.7)	7 (5–8)	0.80
Clinical hepatitis					
Icteric	15 (40.5%)	2 (33.3%)	3 (37.5%)	9 (42.8%)	0.9
Anicteric	22 (59.5%)	4 (66.7%)	5 (62.5%)	12 (57.1%)	
Severe acute hepatitis and/or liver decompensation	13 (36.1%)	3 (50%)	1 (12.5%)	9 (42.8%)	0.2
Symptoms					
Asthenia	18 (48.6%)	3 (50%	4 (50%)	9 (42.8%)	0.9
Fever	12 (32.4%)	1 (16.7%)	2 (25%)	9 (42.8%)	0.4
Nausea/vomiting	15 (40.5%)	2 (33.3%)	4 (50%)	7 (33.3%)	0.7
Diarrhea	4 (10.8%)	1 (16.7%)	1 (12.5%)	2 (9.5%)	0.9
Epigastric pain	13 (35.1%)	3 (50%)	6 (75%)	4 (19%)	0.01
Arthralgia	4 (10.8%)	0	3 (37.5%)	1 (4.8%)	
Extra-hepatic manifestations	11 (29.7%)	3 (50%)	0	7 (33.3%)	0.2
Laboratory parameters at peak					
AST, U/L	1167 (412.5–2088)	1587.5 (613.25–3608.25)	1210.5 (178.25–1912.7)	1228 (284.5–2211.5)	0.7
ALT, U/L	1751 (1.026–2810)	2100.5 (1013–3629.75)	1932.5 (525.5–2930)	1591 (995.5–2736.5)	0.6
Total bilirubin, mg/dL	7 (2.3–12.2)	6.03 (1.4–17.1)	3.5 (1.4–7.2)	10 (2.8–19.2)	0.3
γGT, U/L	265 (164.5–401.5)	437 (168.7–646)	399.5 (202–578.5)	242 (148.5–336.5)	0.1
ALP	317.5 (203.5–577.25)	261 (165.5–539.7)	403 (231.7–661.7)	276 (204–465.7)	0.6
INR	1.17 (1.08–1.5)	1.6 (1.17–2.89)	1.15 (1.03–1.21)	1.15 (1.1–1.5)	0.03

Alanine aminotransferase, ALT; aspartate aminotransferase, ASP; γglutammil transferase, γGT; alkaline phosphatase, ALP; international normalized ratio, INR.

**Table 3 pathogens-09-00832-t003:** Primers used for the amplification of ORF1 and ORF2 of HEV.

	Primer	Location, nt *	Sequence, 5′ -> 3′	Amplicon Length, bp
1 round ORF1	1679	36–56	CCAYCAGTTYATHAAGGCTCC	348 bp
	1680	383–367	TACCAVCGCTGRACRTC	
2 round ORF1	1681	53–71	CTCCTGGCRTYACWACTGC	172 bp
	1682	224–205	GGRTGRTTCCAIARVACYTC	
1 round ORF2	HE44	5934–5912	CAAGGHTGGCGYTCKGTTGAGAC	484 bp
	HE40	6417–6395	CCCTTRTCCTGCTGAGCRTTCTC	
2 round ORF2	HE110	5922–5942	GYTCKGTTGAGACCTCYGGGT	457 bp
	HE41	6378–6356	TTMACWGTCRGCTCGCCATTGGC	

* Nucleotide positions refer to the prototype M73218. Nucleotide, nt; base pair, bp.

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
