# Peer review of "Clinical Characteristics of Acute Hepatitis E and Their Correlation with HEV Genotype 3 Subtypes in Italy"

_pathogens, 2020, doi:10.3390/pathogens9100832_

Round 1

Reviewer 1 Report

Minosse et al. address the question, whether certain HEV GT3 subtypes correlate with distinct clinical phenotypes. In 64 patients the genome were sequenced. Between the varying subtypes no distinct differences in clinical presentation could be shown.

So far, only little information on the clinical impact of HEV GT3 subtypes is available. The paper is well written and the figures well designed.

The major issue is the (retrospective) comparison of subgroups in an overall small sample size (table 2). The retrospective nature of the study and the many comparisons must lead to some statistically significant results. Given the fact, that no significant differences for most categories of the clinical phenotyp (BMI, age, sex,……) could be found, it is very likely that the few significant differences (epigastric pain, INR, viral load) are by chance. This information should be discussed as a limitation, and the differences should be taken with caution. I understand from this manuscript that HEV GT3 subtypes do not cause different clinical phenotypes (or the present study does not prove differences). Please adjust the discussion (page 12, line 288 …..) accordingly.

Comments:

  • The sample numbers should be explained in more detail: 64 sequences, laboratory data of 56 patients, HEV-RNA of 26 patients and clinical data in 37 patients. Although, figure 3 gives an overview, please add the information to e.g., figure 4 (n = x for each GT3 subtype and each age group). Please add also absolute numbers n = x in the text (e.g. page 8 section 2.3.,….)
  • Introduction page 1 line 35: typo? A species? Check
  • Figure 2: I would suggest including the information from 3B into 3A (absolute numbers and %)
  • The decision to include the methods part as section 4 behind results/discussion is odd. It requires lots of scrolling i.e. to look up the definition of severe hepatitis. If not required by the journal that should be changed.
  • The similarity between strains in Europe should be mentioned in the abstract as well as the similarity with a sequence in Italian wastewater. Overall, the abstract does not reflect the manuscript very well. The sentence with 3443 patients is not required. Please avoid overstating the significance of the differences. It should be noted, that no differences could be found for most categories.

Author Response

Reviewer #1

Minosse et al. address the question, whether certain HEV GT3 subtypes correlate with distinct clinical phenotypes. In 64 patients the genome were sequenced. Between the varying subtypes no distinct differences in clinical presentation could be shown.

So far, only little information on the clinical impact of HEV GT3 subtypes is available. The paper is well written and the figures well designed.

  1. The major issue is the (retrospective) comparison of subgroups in an overall small sample size (table 2). The retrospective nature of the study and the many comparisons must lead to some statistically significant results. Given the fact, that no significant differences for most categories of the clinical phenotyp (BMI, age, sex,……) could be found, it is very likely that the few significant differences (epigastric pain, INR, viral load) are by chance. This information should be discussed as a limitation, and the differences should be taken with caution. I understand from this manuscript that HEV GT3 subtypes do not cause different clinical phenotypes (or the present study does not prove differences). Please adjust the discussion (page 12, line 288 …..) accordingly

Response: The following sentence has been added to the Discussion section to address the reviewer’s comments: “This weakened the power of statistical analysis, therefore, categories where statistical significance was observed among GT3 subtypes in our study must be considered with caution.” (Lines 321-322)

Comments:

  1. The sample numbers should be explained in more detail: 64 sequences, laboratory data of 56 patients, HEV-RNA of 26 patients and clinical data in 37 patients. Although, figure 3 gives an overview, please add the information to e.g., figure 4 (n = x for each GT3 subtype and each age group). Please add also absolute numbers n = x in the text (e.g. page 8 section 2.3.,….)

Response: As requested by the reviewer, absolute numbers have been added to the text (Lines 106-107) and in the caption of Figure 4 (Lines 121-127). Figure 3 was also modified.

  1. Introduction page 1 line 35: typo? A species? Check

Response: The sentence has been modified for clarity: “with species A having...” ( Line 37)

  1. Figure 2: I would suggest including the information from 2B into 2A (absolute numbers and %)

Response: We modified Figure 2B, however, absolute numbers have not been included in Figure 2A as we thought that might make the bar graph look cluttered and less clear.

  1. The decision to include the methods part as section 4 behind results/discussion is odd. It requires lots of scrolling i.e. to look up the definition of severe hepatitis. If not required by the journal that should be changed.

Response: The format of the manuscript followed the guidelines provided by the Journal.

  1. The similarity between strains in Europe should be mentioned in the abstract as well as the similarity with a sequence in Italian wastewater. Overall, the abstract does not reflect the manuscript very well. The sentence with 3443 patients is not required. Please avoid overstating the significance of the differences. It should be noted, that no differences could be found for most categories.

Response: The Abstract has been modified accordingly: “GT3c strains were similar to Dutch sequences (96.8%-98.1% identity), GT3e strains showed high similarity (96.8%) with a United Kingdom sequence, while the most related sequences to GT3f Italian strains were isolated in France, Belgium and Japan. One sequence was closely related to another Italian strain isolated in raw sewage in 2016.” (Lines 19-23).

HEV symptoms were comparable among GT3c/e/f patients across most analysed categories (Line 28).

Moreover, the sentenceOf 3443 patients tested for anti-HEV IgM and anti-HEV IgG antibodies, 295 (8.6%) were positive” has been deleted in the abstract.

Reviewer 2 Report

The manuscript under review analyzes the circulating HEV subgenotypes in Lazio. While HEV and its epidemiology are of interest and the paper is in general well written, the study has serious limitations. The low number of tested subjects (and with that the very low statistical power) does not allow for any conclusions regarding the clinical course of infections caused by different subgenotypes.

Methods: The authors state only that 3443 patients were tested for HEV serology. More information is needed. Were they hospitalized patients, patients with hepatitis or screened as donors, or healthy volunteers? Where were they screened?

The authors state that only anti-HEV-IgM patients were tested for HEV RNA. It is known that some patients, mostly immunosuppressed ones that may develop chronic HEV infection, can fail to develop HEV antibodies. How do the authors account for this?

Abstract : “Of 3443 patients tested for 18 anti-HEV IgM and anti-HEV IgG antibodies, 295 (8.6%) were positive”. Positive for IgM, IgG, both or HEV RNA?

I find that the introduction includes a lot widely known information, which is not (all) necessary to understand the study. I would focus more on the subgenotypes, but that is a personal preference.

Author Response

Reviewer #2

The manuscript under review analyzes the circulating HEV subgenotypes in Lazio. While HEV and its epidemiology are of interest and the paper is in general well written, the study has serious limitations. The low number of tested subjects (and with that the very low statistical power) does not allow for any conclusions regarding the clinical course of infections caused by different subgenotypes.

  1. Methods: The authors state only that 3443 patients were tested for HEV serology. More information is needed. Were they hospitalized patients, patients with hepatitis or screened as donors, or healthy volunteers? Where were they screened?

Response: The sentence has been modified for clarification: “Patients included those who were hospitalized and outpatients. All had abnormal ALT levels and were negative for HBV, HAV and HCV  genome and serological markers and genome detection.” (Lines 336-337)

  1. 2. The authors state that only anti-HEV-IgM patients were tested for HEV RNA. It is known that some patients, mostly immunosuppressed ones that may develop chronic HEV infection, can fail to develop HEV antibodies. How do the authors account for this?

Response: We thank the reviewer for the question. We tested for both anti-HEV IgG and IgM in sixteen HIV-positive patients. Test results were negative for both serological markers. Since the HIV-positive patients were immunocompromised, HEV RNA RT-PCR was also performed; however, all results were negative. Since these data were considered superfluous to the focus of the manuscript, they were not included.

  1. Abstract : “Of 3443 patients tested for 18 anti-HEV IgM and anti-HEV IgG antibodies, 295 (8.6%) were positive”. Positive for IgM, IgG, both or HEV RNA?

Answer: We think that “18” in the reviewer’s comment above is a typographical error since the number “18” is not included in the submitted abstract. Nevertheless, this sentence has been deleted in the revised version of the manuscript (suggestion of referee #1). It is stated in the results section of the manuscript that 3443 patients were tested for an anti-HEV IgM response.

  1. 4. I find that the introduction includes a lot widely known information, which is not (all) necessary to understand the study. I would focus more on the subgenotypes, but that is a personal preference.

Answer: We prefer to include an overview of the HEV genotypes and their prevalence worldwide and in Europe to better explain the goal of our study. The description of subtypes and their clinical manifestations are placed in the Discussion section for an easier comparison of our results to those reported in literature.

We added the ethical approval code, as requested by Ms Ljubica Valaga (assistant Editor, MDPI AG). (Lines 348-349)

Reviewer 3 Report

Dr. Minosse and colleagues presented an obviously revised manuscript showing an interesting observational study in which they aimed to assess clinical characteristic of acute hepatitis E and a possible correlation to HEV-3 subtypes in patients from Italy. To show this the colleagues retrospectively analysed 3443 patient data and samples between 2011 and 2019. 8.6% were HEV IgM positive while 88 samples showed HEV-RNA positivity. Analyses of HEV geno- and subtypes showed not unexpected a predominance of HEV-3f followed by HEV-3e and HEV-3c. HEV RNA loads and clinical manifestations revealed no correlation between HEV subtypes except for epigastric pain. However, this could also be by chance due to the low sample number. From their findings the colleagues conclude that there is no correlation between severity and course of acute hepatitis E and HEV-3 subtypes.

Overall, the manuscript is well performed and concise in its content showing convincing data. The number of patient samples analysed (3443) is adequate; however, the number HEV positive samples 265 divided in subgroups that could further be analysed is small. The figures and tables are well constructed. Actually there is no major criticism with regard to the content and sufficiency of the manuscript. However, there are some minor limitations which should be addressed.

Comments:

  • Introduction section, line 39 and 47 and throughout the text. To be politically correct please change “developing countries” to “low income countries” or “countries with limited resources”.
  • Result section, lines 80 ff. It is not clear to me whether the 295 positive tested samples were IgM or IgG positive.
  • The 88 HEV RNA positive samples were also IgM positive? And the 256 available samples where from which collective? From the 295? This part is confusing, please clarify.
  • Figure 3. The numbers in the figure correlates to which data in the text? Please, explain in more detail in the text or the figure legend.

Author Response

Reviewer #3

Comments:

1) Introduction section, line 39 and 47 and throughout the text. To be

politically correct please change “developing countries” to “low income

countries” or “countries with limited resources”.

Response- the text had been modified as suggested by the referee (line 39, 47)

2) Result section, lines 80 ff. It is not clear to me whether the 295

positive tested samples were IgM or IgG positive.

Response-the sentence has been modified  : “Of 3443 patients tested for anti-HEV IgM and IgG during 2011-2019, 295 (8.6%, 295/3443) were positive for both serological markers” (Line 82)

3) The 88 HEV RNA positive samples were also IgM positive? And the 256

available samples where from which collective? From the 295? This part is

confusing, please clarify.

Response-Yes.The sentence had been modified for clarity: “HEV RNA was detected in 88 (34.4%) of 256 available  anti-HEV IgG and IgM positive patient-derived samples (Figure 1)” ( Lines 83, 84) . HEV RNA was not tested in 39 patients because of the samples were not available. This information is reported in Figure 1.

4) Figure 3. The numbers in the figure correlates to which data in the text?

Please, explain in more detail in the text or the figure legend.

Response – Thank for this observation. We explained the figure’s numbers in the legend.

The analysis were carried out on sixty-four GT3 strains that were underwent to phylogenetic characterization. Moreover, we described more in detail the median ALT, AST, Bilirubin calculation in the text (Lines 107,109), and Figure 4 legend  had been modified.

Round 2

Reviewer 2 Report

My comments have been partly addressed.

Author Response

no comments for this reviewer